# Intravenous Fosfomycin: A Potential Good Partner for Cefiderocol. Clinical Experience and Considerations

**DOI:** 10.3390/antibiotics12010049

**Published:** 2022-12-28

**Authors:** Andrea Marino, Stefano Stracquadanio, Edoardo Campanella, Antonio Munafò, Maria Gussio, Manuela Ceccarelli, Renato Bernardini, Giuseppe Nunnari, Bruno Cacopardo

**Affiliations:** 1Department of Biomedical and Biotechnological Sciences, University of Catania, 95123 Catania, Italy; 2Unit of Infectious Diseases, Department of Clinical and Experimental Medicine, ARNAS, Garibaldi Hospital, University of Catania, 95123 Catania, Italy; 3Section of Pharmacology, Department of Biomedical and Biotechnological Sciences, University of Catania, 95123 Catania, Italy; 4Unit of Infectious Diseases, Department of Clinical and Experimental Medicine, University of Messina, 98124 Messina, Italy

**Keywords:** cefiderocol, novel antimicrobial strategies, intravenous fosfomycin, *Pseudomonas aeruginosa*, *Acinetobacter baumannii*, MDR Gram negative bacteria

## Abstract

Multidrug resistant Gram-negative bacteremia represents a therapeutic challenge clinicians have to deal with. This concern becomes more difficult when causing germs are represented by carbapenem resistant *Acinetobacter baumannii* or difficult-to-treat *Pseudomonas aeruginosa*. Few antibiotics are available against these cumbersome bacteria, although literature data are not conclusive, especially for *Acinetobacter*. Cefiderocol could represent a valid antibiotic choice, being a molecule with an innovative mechanism of action capable of overcoming common resistance pathways, whereas intravenous fosfomycin may be an appropriate partner either enhancing cefiderocol activity or avoiding resistance development. Here we report two patients with MDR Gram negative bacteremia who were successfully treated with a cefiderocol/fosfomycin combination.

## 1. Introduction

Worldwide, bacterial bloodstream infections (BSI) are associated with substantial morbidity and mortality [1,2,3], becoming devastating when BSI are caused by multidrug-resistant (MDR) organisms, such as methicillin-resistant *Staphylococcus aureus* (MRSA), vancomycin-resistant *Enterococcus* spp. (VRE), and MDR Gram-negative bacteria (MDR-GNB), especially difficult-to-treat *Pseudomonas aeruginosa* (DTT-Pa) and carbapenem-resistant *Acinetobacter baumannii* (CRAB) [4,5,6,7,8].

Suspecting MDR-GNB BSI, empiric combination therapy rather than monotherapy is a common option allowing an increase of the spectrum of antibiotic activity, to achieve faster bacterial clearance, to assure possible synergistic effect and to avoid—or at least to reduce—the onset of bacterial resistance [9,10,11,12]. Marking this point, Schmid et al. reported reduced mortality rates in patients with infections caused by carbapenemase-producing *Enterobacterales* (especially BSI) treated with combination regimens, when compared with those treated with monotherapy [9].

Recently, both new antibiotics—such as ceftolozane/tazobactam, ceftazidime/avibactam, meropenem/vaborbactam [13,14] and cefiderocol—and old molecules—such as intravenous (IV) fosfomycin [15,16] and colistin—provided efficient treatment options to counteract severe infections caused by MDR organisms. Furthermore, ceftazidime/avibactam, ceftolozane/tazobactam, cefiderocol [17] and intravenous (IV) fosfomycin [18] represent potential carbapenem sparing alternatives as shown by Xie et al. by determining the number of carbapenem-days saved using the aforementioned options [19].

Cefiderocol is an injectable cephalosporin acting with “Trojan horse”-like mechanism by using active iron transporters. This siderophore antibiotic has a potent broad-spectrum activity against aerobic GNB, including MDR Enterobacterales, as well as CRAB and DTT-Pa [20,21]. Cefiderocol is considered resistant to several β-lactamases, also thank to its novel mechanism of cell-entry as siderophore molecules mimicking. In addition, cefiderocol overcomes other common mechanisms of β-lactam resistance among Gram-negative bacteria, including porin deficiency and efflux pump up-regulation [11].

On the other hand, fosfomycin, the sole antibiotic of the epoxide group, which acts by inhibition of bacterial wall formation [22], is a broad-spectrum bactericidal antibiotic available in three forms: fosfomycin tromethamine (a soluble salt), fosfomycin calcium for oral use [23] and fosfomycin disodium for intravenous use [15]. Fosfomycin is effective against several resistant Gram-positive and Gram-negative bacteria, especially as “partner-drug” of other active molecules. Despite the general belief regarding *A. baumannii* intrinsic fosfomycin resistance, limited information is available in scientific literature concerning the involved mechanisms [24,25]. Recently, the activity of IV fosfomycin-based combination regimen (together with antibiotics, such as aminoglycosides, colistin and minocycline) against MDR *A. baumannii* has renewed interest in fosfomycin as an attractive treatment partner [12,26,27].

As far as we know, only few reports described data about cefiderocol and IV fosfomycin combination for the treatment of GNB infections (including BSI) [11,28,29,30], whereas there is important scientific literature data about fosfomycin and other antibiotics association [12,26,31,32,33].

Herein, we describe two cases of MDR GNB BSI treated with a combination of cefiderocol and IV fosfomycin. We also discussed our choice in light of literature evidence about IV fosfomycin combination with cefiderocol.

## 2. Case Presentation

### 2.1. Case 1

A 75-year-old woman with a medical history of radical cystectomy and right nephrectomy due to urothelial tumor, unilateral cutaneous ureterostomy and chronic renal failure. She was brought to the emergency department seven days after the substitution of ureterostomies’ tutor due to the onset of high fever (up to 39 °C) and chills associated with hematuria. At admission, blood pressure was 110/60 mmHg; respiratory rate was 18/min; and GCS was 13 (qSOFA score was 1).

Two sets of blood cultures and urine cultures were performed, and empiric antibiotic therapy was started with piperacillin/tazobactam, dosed accordingly to patient’s impaired renal function. 

Blood tests showed high white blood cell count along with mild anemia and reduced platelet levels (Table 1). Inflammatory markers were elevated, as well as procalcitonin levels. Urinalysis revealed bacteriuria and leukocyturia. Abdomen ultrasound and urinary CT did not show any focal sign of infection.

All sets of blood cultures and urine cultures resulted positive for *Pseudomonas aeruginosa* carbapenem resistant (resistance pattern was assessed with BD Phoenix with exception of fosfomycin, colistin and cefiderocol. See Table 2 for the full antibiogram). Fosfomycin susceptibility was confirmed by commercial AD fosfomycin agar dilution test (cat. no. 77061, Liofilchem, Italy), giving a minimum inhibitory concentration (MIC) value of 16 mg/L. The strain was susceptible to colistin—tested by broth microdilution (colistin sulfate salt, cat. no. C4461, Sigma-Aldrich, St. Louis, MO, USA). After cefiderocol susceptibility testing performed through disk diffusion (cat. no. 9266, Liofilchem, Italy) [34], antibiotic therapy was switched to IV cefiderocol 1 g three times daily plus IV fosfomycin 4 g two times daily, based on her renal clearance (eGFR of 26.2 mL/min, serum creatinine 1.85 mg/dL). Furthermore, the patient’s ureterostomy was substituted.

Within 72 h following targeted antibiotic treatment, the fever disappeared; inflammatory markers started decreasing; and blood cultures taken 48 h apart tested negative. 

Antibiotic regimen was administered for 10 days, achieving clinical cure and microbiological eradication, assessed with two negative blood cultures at the end of therapy. The patient was successfully discharged after seven days after the end of the therapy, and no infection relapse occurred during this follow-up. 

### 2.2. Case 2

A 45-year-old man with Wilson disease complicated by cirrhosis and esophageal varices along with chronic renal failure was admitted to the emergency department due to hematemesis, treated with endoscopic variceal ligation combined along with terlipressin administration. 

On the 5th day from the time of admission, due to the onset of fever (up to 38.5 °C), two sets of blood cultures were performed, and empiric antibiotic therapy with piperacillin/tazobactam was started. His blood pressure was 120/70 mmHg; respiratory rate was 20/min; and GCS was 15 Patient’s (qSOFA was 0).

Blood tests showed elevated white blood cell count along with higher CRP, ESR and procalcitonin levels (Table 1).

*Acinetobacter baumannii* only susceptible to colistin—tested by broth microdilution (colistin sulfate salt, cat. no. C4461, Sigma-Aldrich, St. Louis, MO, USA)—and cefiderocol—tested by disk diffusion (cat. no. 9266, Liofilchem, Italy)—[34] (resistance pattern was assessed with BD Phoenix with exception of fosfomycin, colistin and cefiderocol. See Table 3 for the full antibiogram) was recovered from blood cultures. Fosfomycin MIC was evaluated as aforementioned and resulted 64 mg/L. Based on his renal clearance, antibiotic therapy was switched to IV cefiderocol 1.5 g 3 times daily plus IV fosfomycin 4 g 3 times daily.

Within 72 h following cefiderocol based regimen, the fever disappeared; inflammatory markers decreased; and blood cultures taken 48 and 72 h apart tested negative.

Antibiotic therapy was administered for 7 days. After 5 days, the patient was successfully transferred to the hepatology unit for liver follow-up.

## 3. Discussion

The treatment of MDR-GNB infections represents a clinical and therapeutic challenge [7,13,35,36,37], with several obstacles to overcome: rapid diagnostic testing, infection control measures and prompt starting of an effective treatment following appropriate empirical coverage. 

Moreover, to wisely guide the choice and management of antibiotic regimen, it is mandatory to obtain the latest information about local microbiological epidemiology [38]. The patients we presented were affected by multiple severe comorbidities complicated by BSI due to MDR-GNB, i.e., DTT-Pa and CRAB. Therapeutic options were limited due to both antimicrobial resistance patterns and patients’ characteristics, especially renal clearance. 

On the basis of antibiotic susceptibility tests along with patients’ challenging clinical conditions, both our patients were treated with cefiderocol plus IV fosfomycin, achieving microbiological eradication and clinical cure. 

In detail, as regarding the first case, we chose to administer cefiderocol-based therapy due to the shortage of ceftolozane/tazobactam provisions in our center and due to ceftazidime/avibactam high MIC (8 mg/L), which was near to the clinical breakpoint, in order to avoid either bacterial resistance or treatment failure, which are common features of severe infections involving fragile patients. Due to its high rate of nephrotoxicity (20–60% in different studies), colistin based therapy was not taken into consideration. 

Considering the second case, due to both the limited antibiotic choice because of *Acinetobacter* resistance profile and the patient’s comorbidity, especially renal impairment, we decided again to avoid colistin regimen switching to cefiderocol plus fosfomycin.

Although it has no antimicrobial activity against Gram-positive and anaerobic germs, cefiderocol, the newest siderophore antibiotic, revealed significant antibacterial activity towards MDR GNB, including non-fermenting bacilli—such as *Acinetobacter* or *Stenotrophomonas*—and Enterobacterales—such as *Klebsiella* [39,40,41] thanks to its unique pharmacodynamic which assure bypassing common bacterial resistance mechanisms.

Although the use of a β-lactam/β-lactamases-inhibitor combination (BLIC) could represent a valid alternative for the treatment of DTT-Pa and CRAB, the choice should be done on the basis of molecular resistance mechanisms, not always investigated in hospital settings. Cefiderocol has the benefit to overcome the known β-lactamases-based mechanism of resistance—especially against metallo β-lactamases (MBL)—and, thus, could be a feasible option. Moreover, there are not randomized studies investigating the different efficacy of cefiderocol compared to ceftazidime-avibactam, ceftolozane-tazobactam or imipenem-cilastatin-relebactam in the treatment of DTT-Pa or *Acinetobacter*. Eventually, fosfomycin addition showed a significant efficacy even against *A. baumannii* strains, as its peculiar mechanism of action seems to evoke a stress, even in resistant strains as usually are *Acinetobacter baumannii* isolates [42], that make bacteria more susceptible to other molecules.

Indeed, cefiderocol, a catechol-type siderophore with a cephalosporin core and side chains similar to cefepime and ceftazidime, is able to overcome cumbersome bacterial resistance mechanisms, including the production of β-lactamases, even metallo-beta-lactamases (MBL), up-regulation of efflux pump expression and porin deficiency. 

CREDIBLE-CR and APEKS-NP studies demonstrated cefiderocol non-inferiority when compared to the best available therapy to treat either cUTIs or nosocomial Gram-negative pneumonia [39,43].

In a recent retrospective study, Pascale et al. assessed cefiderocol monotherapy efficacy in MDR *A. baumannii* infections compared to colistin, showing no differences in all-cause mortality rate [44]. Similar results were obtained by Falcone et al. analyzing a population, including MBL producing *Enterobacterales* and non-fermenting bacteria such as *A. baumannii* and *S. maltophilia* [28].

Some authors suggest adding a second agent to cefiderocol in order to avoid resistance development and therapeutic failure, especially considering critically ill patients and those with limited therapeutic options [45,46]. A recent sectional survey by Lupia et al. [47] reported that most clinicians use IV fosfomycin as a common cefiderocol partner drug in the treatment of both *Pseudomonas* and *Acinetobacter* infections. 

Fosfomycin, thanks to its favorable PK/PD profile, has been recognized as a good antibiotic option for the treatment of systemic and deep-seated infections. Indeed, after intravenous administration, fosfomycin results in a sufficient drug concentration at different body sites [48]. Recently, Antonello et al. performed a systematic review about fosfomycin’s synergistic properties, underlying the suitable features of this molecule as partner drug alongside the nearly total absence of antagonisms towards other drugs. Furthermore, authors showed that fosfomycin-based regimens are characterized by stronger bactericidal effect toward *P. aeruginosa* with significant synergistic interactions when associated with chloramphenicol, aminoglycosides or cephalosporins, as well as against *Acinetobacter* spp. especially together with sulbactam and penicillins [31].

Although there is not conclusive evidence about combination therapy over monotherapy, there are scientific reports which state that monotherapy represents an independent predictor of 28-day mortality (as was absence of infectious diseases specialist consultation) compared with combination therapy based mostly on fosfomycin [49].

Moreover, Bavaro et al. reported cefiderocol combination regimen to treat DTT *P. aeruginosa*, choosing fosfomycin as second agent due to its ability to reduce cefiderocol MIC [30]. 

However, it is not clear if fosfomycin exhibits a time- or a concentration-dependent bactericidal effect [50,51]; therefore, some authors assert that it might depend on the microorganism [52]. 

Furthermore, the new interest behind the use of IV fosfomycin in the treatment of infections due to GNB implies the need of adequate susceptibility testing of this antibiotic. Despite agar dilution (AD) is the reference method recommended by the European Committee on Antimicrobial Susceptibility Testing (EUCAST) for the detection of fosfomycin susceptibility in GNB; it is not suitable for every hospital setting because it is a laborious and time-consuming process. In the last years, various commercial automated systems have been developed to detect antibiotic susceptibility faster and practically. Studies have shown high categorical agreement of these methods with AD determining the susceptibility of fosfomycin [53,54], but as reported by EUCAST, the accuracy of the results depend on the method used for the isolation’s microorganisms and on adherence to the manufacturers’ instructions [55,56].

Likewise, cefiderocol susceptibility testing is cumbersome due to the need of iron depleted broth, the area of technical uncertainty (ATU) of the disk diffusion method and the lack of validate e-test strips suitable for the main MDR GNB, except for *P. aeruginosa*. Furthermore, cefiderocol in vitro activity against *A. baumannii* is still under a magnifier [41,57].

Due to recent introduction of cefiderocol in the clinical practice, to date only few data have been published about the best combination therapy for the management of MDR infections, and both its use as monotherapy and its clinical impact are still debated. Nonetheless, fosfomycin—thanks to its peculiar mechanism of action—is recognized as one of the best companions for combination therapy [22,58,59,60,61,62,63,64,65,66,67]; indeed, Gatti and colleagues have already proposed an algorithm for targeted therapy of infection caused by *P. aeruginosa*, suggesting the association of cefiderocol and fosfomycin, especially for patients in intensive care units [68]. 

The fosfomycin definition shifted from “intrinsically inactive” to a “miscellaneous agent” to treat CRAB infections [27], as described clinically by Bavaro et al. [30] and in vitro by Nwabor et al. [27]; the latter demonstrated the potency of fosfomycin combination with other antibiotics against CRAB, in terms of MIC reduction and restitution of efficacy. Although we reported only one patient with CRAB infection and currently available data are insufficient for substantial conclusions, we administered fosfomycin as partner drug even if *Acinetobacter* was resistant, achieving successful clinical outcome.

## 4. Conclusions

Our experience suggests that a combination of intravenous fosfomycin and cefiderocol may represent a valid therapeutic option to treat GNB BSI, both as a carbapenem sparing strategy and as a treatment option in difficult to treat infections. Indeed, this combination should be carefully evaluated in larger studies and prudently compared to other available options to better assess its clinical efficacy, microbiological eradication rate and its ability to prevent antibiotic resistance development.

Intravenous fosfomycin as a partner drug in an antibiotic regimen containing new available molecules seems to be a reasonable option in view of its favorable PK/PD and its synergistic effects with several drugs. Further and stronger studies are needed both to assess whether monotherapy would be more effective and safer than combination therapy and to evaluate if a partner drug, such as fosfomycin, could “protect” newer antibiotics, such as cefiderocol, against bacterial resistance development. Therapeutic drug monitoring (TDM) may be a possible solution to assess antibiotic efficacy in frail patients, such as those with impaired renal clearance [69].

## Figures and Tables

**Table 1 antibiotics-12-00049-t001:** Laboratory findings at admission and after antibiotic treatment. WBC: White blood cells; AST: aspartate aminotransferase; ALT: alanine aminotransferase; LDH: lactic dehydrogenase; eGFR: estimated glomerular filtration rate; CRP: C-reactive protein; ESR: erythrosedimentation rate.

	Patient 1		Patient 2	
Laboratory findings, unit (reference range)	Admission	End of therapy	Admission	End of therapy
WBC, cells/mmc (4000–10,000)	14,500	4200	13,200	3700
Neutrophils, % (40–75)	82.3	48.3	88.2	58.5
Lymphocytes, % (25–50)	12.6	42.9	6.1	28
Monocytes, % (2–10)	4.7	7.1	3.3	5.5
Platelets, cells/mmc ×10^3^ (150–400)	114	118	24	46
Haemoglobin, g/dL (12–16)	8.2	9.1	7.9	8.2
AST, UI/L (15–35)	17	45	14	31
ALT, UI/L (15–35)	6	16	9	45
LDH, UI/L (80–250)	174	184	431	142
Creatinine, mg/dL (0.8–1.2)	1.98	1.74	1.78	0.65
e-GFR EPI-CKD	24.1	28.2	45.4	118.3
CRP, mg/dL (0–0.5)	13.07	0.51	6.44	0.65
ESR, mm/h (0–10)	123	63	35	6
Procalcitonin, ng/mL (<0.05)	4	0.07	8.67	0.12
D-dimer, ng/mL (<250)	1207	763	1715	644

**Table 2 antibiotics-12-00049-t002:** *Pseudomonas aeruginosa* antibiotic susceptibility. MIC: minimum inhibitory concentration; S: susceptible; R: resistant.

Antibiotics	MIC (mg/L)	S/R	AST
Amikacin	<8	S	As given by BD Phoenix
Cefepime	>8	R	
Ceftazidime	>8	R	
Ciprofloxacin	2	R	
Gentamicin	4	S	
Imipenem	4	R	
Levofloxacin	2	R	
Meropenem	>8	R	
Piperacillin/tazobactam	>16	R	
Ceftazidime/avibactam	8	S	
Fosfomycin	16	S	AD
Colistin	0.5	S	BMD
Cefiderocol	13 mm	S	DD

AST: antimicrobial susceptibility tests; AD: agar dilution; BMD: broth microdilution; DD: disk diffusion.

**Table 3 antibiotics-12-00049-t003:** *Acinetobacter baumannii* antibiotic susceptibility. MIC: minimum inhibitory concentration; S: susceptible; R: resistant.

Antibiotics	MIC (mg/L)	S/R	AST
Amikacin	>16	R	As given by BD Phoenix
Cefepime	>8	R	
Cefotaxime	>16	R	
Ceftazidime	>8	R	
Gentamicin	>4	R	
Imipenem	>8	R	
Levofloxacin	>1	R	
Meropenem	>8	R	
Piperacillin/tazobactam	>16	R	
Tetracyclin	>8	R	
Trimethoprim/sulfametoxazole	>4/76	R	
Fosfomycin	128	N/A	AD
Colistin	1	S	BMD
Cefiderocol	15 mm	S	DD

AST: antimicrobial susceptibility tests; AD: agar dilution; BMD: broth microdilution; DD: disk diffusion; NA: not applicable.

## Data Availability

Not applicable.

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
