# Peer review of "Intravenous Fosfomycin: A Potential Good Partner for Cefiderocol. Clinical Experience and Considerations"

_antibiotics, 2022, doi:10.3390/antibiotics12010049_

Round 1

Reviewer 1 Report

I would like to thank the authors for this nice paper. I have some suggestions and concerns to be addressed.

Title:

I suggest not to phrase the title as a question

Can we consider this as case series given only 2 patients?

Introduction:

Line 55: the sentence is not clear, please rephrase

Case 1:

For how long this patient was followed after the end of treatment? No relapse?

Why did you decide to use this combination?

What about ceftolozane/tazobactam and ceftazidime/avibactam? Did you test them? Why did not you use them?

Why did not you include cefiderocol susceptibility in table 2?

What about colistin susceptibility and MIC?

Case 2:

It could have been possible that clinical cure occurred if patient received cefiderocol alone. How can we make sure that fosfomycin can add potential benefit?

Again, did you follow this patient and assess relapse?

Cefiderocol and colistin susceptibility need to be in table 2

You mentioned you did literature review; however, these are part of your discussion. We do not see a specific table or specific paragraph of the previous studies. I suggest you either do a specific table to summarize the previous data or remove this part from your title and the aim of the study.

We should know what is the place of therapy of this combination given that we have several novel agents (specially for pseudomonas) and specially that IV Fosfomycin is not available in the US and several other countries.

Reviewer 2 Report

The subject of the paper is important and interesting. However its content is not as good as the subject itself. Major revision is required

Round 2

Reviewer 2 Report

The authors have improved the manuscript. The conclusions are now correct.

However, tables 2 and 3 still require significant improvement.

Author Response

Dear reviewer,

Thank you for your valuable revision and suggestions.

We changed the tables as you suggested, fixing what you pointed out.

With regard to Pseudomonas aeruginosa resistance typing, although Phoenix system mentioned the possible presence of carbapenemases without specifying the actual mechanism, we did not perform supplementary tests at that moment. 

We absolutely agree with you regarding the clinical and epidemiological benefit of knowing resistance pattern/mechanisms for this type of infections and we will take it into account for our next studies. 

Kind regards,

The Authors

Round 3

Reviewer 2 Report

The work up is quite complete, only one minor correction should be done. Delete trimethoprim-sulfamethoxazole from table 2 - P. aeruginosa is also intrinsic resistant to this antibiotic.

Author Response

Sorry, we forgot to do it. The table is now updated. 

Many thanks